# Genetic Characterization of the Partial Disease Resistance of Rice to Bacterial Panicle Blight and Sheath Blight by Combined QTL Linkage and QTL-seq Analyses

**DOI:** 10.3390/plants12030559

**Published:** 2023-01-26

**Authors:** John Christian Ontoy, Bishnu Shrestha, Hari Sharan Karki, Inderjit Barphagha, Brijesh Angira, Adam Famoso, Jong Hyun Ham

**Affiliations:** 1Department of Plant Pathology and Crop Physiology, LSU AgCenter, Baton Rouge, LA 70803, USA; 2Department of Plant Pathology and Crop Physiology, College of Agriculture, Louisiana State University, Baton Rouge, LA 70803, USA; 3H. Rouse Caffey Rice Research Station, LSU AgCenter, Rayne, LA 70578, USA.

**Keywords:** *Oryza sativa*, *Burkholderia glumae*, *Rhizoctonia solani*, host resistance, RILs, QTL, linkage analysis, QTL-seq

## Abstract

Bacterial panicle blight (BPB) and sheath blight (SB) are major diseases of rice and few cultivars have shown a high level of resistance to these diseases. A recombinant inbred line (RIL) population developed from the U.S. cultivars Jupiter (moderately resistant) and Trenasse (susceptible) was investigated to identify loci associated with the partial disease resistance to BPB and SB. Disease phenotypes in BPB and SB, as well as the days-to-heading (DTH) trait, were evaluated in the field. DTH was correlated to BPB and SB diseases, while BPB was positively correlated to SB in the field trials with this RIL population. Genotyping was performed using Kompetitive Allele Specific PCR (KASP) assays and whole-genome sequence (WGS) analyses. Quantitative trait locus (QTL) mapping and bulk segregant analysis using a set of WGS data (QTL-seq) detected a major QTL on the upper arm of chromosome 3 for BPB, SB, and DTH traits within the 1.0–1.9 Mb position. Additional QTLs associated with BPB and SB were also identified from other chromosomes by the QTL-seq analysis. The QTLs identified in this study contain at least nine candidate genes that are predicted to have biological functions in defense or flowering. These findings provide an insight into the complex nature of the quantitative resistance to BPB and SB, which may also be closely linked to the flowering trait.

## 1. Introduction

Bacterial panicle blight (BPB) [1,2] and sheath blight (SB) [3,4] are major diseases in rice-growing areas in the United States. BPB occurs sporadically in the rice-growing states of the southern U.S., which include Arkansas, Louisiana, Mississippi, and Texas [5]. However, this disease can be epidemic, extensively causing severe damage in a conducive weather condition. A survey reported that the disease was observed in approximately 60% of Louisiana rice fields when the disease outbreak was severe [6]. Two species of the bacterial genus *Burkholderia, B. glumae* (Kurita and Tabei 1967) [7] and *B. gladioli* (Severini 1913) [8], are known to cause BPB [5], but *B. glumae* is considered as the major causal agent because it is more prevalent and aggressive than *B. gladioli* [9]. *B. glumae* (Kurita and Tabei 1967) [7] (formerly *Pseudomonas glumae* Kurita and Tabei 1967) was first reported in Japan (Kurita and Tabei 1967) and became widely spread to East Asia [10], Latin America [11,12] South Africa [13], and the United States [5]. Especially, yield losses from BPB have been a chronic limiting factor for stable rice production in the southern United States [9]. The bacteria grow well at the optimum temperature range, 30 to 35 C, becoming prevalent in tropical and semi-tropical environments [9]. A recent study reported that warming of temperature increases the outbreaks of BPB, causing negative impacts on U.S. rice production [14]. Sheath blight (SB), caused by the soilborne fungus *Rhizoctonia solani* Kuhn, poses a major threat to the grain rice yield and quality worldwide [15]. The disease was first reported in Japan in 1910 and subsequently reported worldwide [16]. The fungus is semi-saprophytic with a broad host range [17]. In the United States, highly intensive production systems can incur a yield loss of up to 50% due to SB [18].

Effective management of BPB and SB is critical to maintain stable rice production and minimize yield losses. Utilizing host resistance is the most sustainable and practical approach to manage plant diseases and can be complemented by additional disease management practices. Unfortunately, few options for disease management are currently available for BPB. There has been no major gene or locus detected for rice resistance to BPB [19]. Although some QTLs have been identified from different studies [20,21], genes responsible for those QTLs have not been known. Only a limited number of cultivars with partial resistance are available and no chemical control measure is available in the U.S. Although oxolinic acid has been used for BPB control in other countries [22], this antibiotic is not labeled for use on rice in the U.S. [23]. Due to the absence of highly effective pesticides and increasing outbreaks of BPB, it is imperative to develop BPB-resistant cultivars [24]. In the U.S. alone, more than 200 elite breeding lines and cultivars were evaluated for BPB [23]. Results of multiple years of evaluations revealed that most of the U.S. rice cultivars are highly susceptible to BPB and there is no promising resistant line available yet [25].

Meanwhile, many studies have been performed to understand and enhance the resistance to SB, which is controlled by multiple QTLs [26,27,28,29]. Nevertheless, it has been difficult to validate these QTLs because of a limited amount of agreement between QTLs identified by different research groups [30], and few major SB resistance genes have been identified from cultivated or wild rice [31]. Due to this situation, breeding practice using marker-assisted selection (MAS) to enhance SB resistance in commercial rice cultivars has not been reported much [32], except that a few fine-mapped loci, such as *qSB-9TQ* [33] and *qSB-11LE* [34], have been applied for breeding.

Quantitative disease resistance (QDR) is generally conferred by multiple loci and considered to be durable, leading to a partial rather than complete suppression of disease [35,36]. An effective approach for studying complex and polygenic forms of phenotype, such as QDR, is quantitative trait locus (QTL) mapping, which involves testing of DNA markers throughout a genome for the likelihood of the presence of QTL [36,37]. Recently, application of whole-genome sequencing (WGS) to plant genetic studies has generated sufficient data for the identification of trait-associated genomic loci or genes. Particularly, combining the bulked-segregant analysis (BSA) with the WGS data of two bulked groups representing opposite phenotypic traits in a population has paved the way to the development of the genetic analysis tool called QTL-seq [38]. QTL-seq analysis allows rapid identification of QTLs and candidate genes without need for DNA markers for genotyping, which is the most time-consuming and costly procedure needed for the conventional QTL analysis [39].

In this study, we identified and investigated the QTLs associated with the QDR of rice to BPB and SB using two approaches, QTL linkage analysis and QTL-seq analysis, with a biparental RIL population. The donor parent, Jupiter, has been known to be moderately resistant to both BPB and SB [40]. Hence, it was important to determine common and unique genomic regions that confer the resistance to those diseases. The disease-resistant recombinant inbred lines (RILs) and the QTLs identified in this study will facilitate the development of BPB- and SB-resistant cultivars in the future and will help to elucidate the underlying mechanism of rice resistance to these important rice diseases.

## 2. Results

### 2.1. Phenotypic Variations within the RIL population

The RILs and the parents (cvs. Jupiter and Trenasse) exhibited variable phenotypes in BPB, SB, and days to 50% heading (DTH) (Table 1). The parents, Jupiter and Trenasse, differed significantly in BPB and SB traits, where Jupiter recorded lower values consistently in the BPB and SB scores, as well as DTH values, for multiple years compared to Trenasse. Photo images that represent the disease phenotypes of Jupiter and Trenasse are presented in Appendix A. We also observed significant variations among the 286 RILs derived from Jupiter and Trenasse in all three traits. The mean values of the RILs for BPB, SB, and DTH were all in between the mean values of the two parents (Table 1). However, all the traits were not normally distributed within the RIL population as the basic assumption used for QTL mapping, and the presence of transgressive segregation in both directions was found (Figure 1). The large shift of DTH between 2012 and 2013 seasons was likely due to different environmental conditions (especially weather conditions) in the field. In addition, there was a wide range of variation in heritability values (h^2^) among the three traits evaluated, in which the values ranged from as low as ~0.24 for BPB (and ~0.49 for SB) to ~0.71 for DTH (Table 1). The higher h2 value for the DTH trait indicates its stronger link to genetics compared to those for the disease traits.

### 2.2. Correlations among the Traits of BPB, SB, and Days to 50% Heading (DTH)

The Pearson’s correlation coefficients indicated that there were significant correlations among the BPB, SB, and DTH traits (Table 2). The SB disease scores of both 2012 and 2014 showed significant negative correlations with DTH. Likewise, the correlations between the BPB disease scores and the DTH data were significantly negative in both 2012 and 2013 (Table 2). It was also observed that the BPB disease scores was positively correlated to the SB scores, although a significant correlation was obtained only between the SB score of 2012 and the BPB score of 2013 (Table 2). These trends mean that early-heading lines tend to show more severe symptoms of BPB and SB diseases compared to late-heading lines, and the lines showing more resistance to BPB also tend to be more resistant to SB. The disease phenotypes for BPB and SB also correlated with plant height, except that the BPB phenotype in 2013 did not show a significant correlation with the plant height phenotype in the same year (Appendix A).

### 2.3. QTL Analysis of the Traits for BPB, SB, and DTH

QTL analyses were conducted with two-season data for SB (2012 and 2014), BPB (2012 and 2013), and DTH (2012 and 2013) for the Trenasse/Jupiter RIL population using inclusive composite interval mapping (ICIM). A total of thirteen QTLs were detected for the three traits by ICIM from the phenotype data of all years, and five of them were overlapped between the markers SNP225 and SNP91 (Table 3 and Figure 2).

Three QTLs for SB (i.e., qSB2.1, qSB3.1, and qSB9.1) were detected by ICIM with significant additive effects. One major QTL was mapped on chromosome 3, while one minor QTL each on chromosomes 2 and 9. A minor QTL was defined as a QTL explaining less than 10.0% of the phenotypic variation, whereas a major QTL was defined as a QTL with ≥10%. All SB-related QTLs have phenotypic variance ranging from 4.32% to 26.34%, among which the most significant locus, qSB3.1, was detected in both 2012 and 2014 (Table 3). The Trenasse alleles had an increasing effect on the SB score index as indicated by the positive additive values in all of the detected SB-related QTLs, suggesting the desirability of the Jupiter alleles in the QTLs for SB (Table 3). In the ICIM analysis of QTLs for BPB, seven QTLs were detected with significant additive effects (Table 3). One major QTL was mapped on chromosome 3, which was in the same chromosomal location for the major QTL for SB, while there was one minor QTL each on chromosomes 1, 2, 3, 4, 5, and 11 (Figure 2). The BPB-related QTLs have phenotypic variance ranging from 3.26% to 19.82% (Table 3). Like in the QTLs for SB, the Trenasse alleles had increasing effects in the majority (five out of seven) of the BPB-related QTLs detected, including the major QTL qBPB3.1, also suggesting the desirability of the Jupiter alleles in the BPB QTLs except for qBPB4.1 and qBPB11.1 (Table 3). For DTH, one major QTL (qDTH3.1) was also mapped on chromosome 3, which was in the same location of the major QTLs for BPB (qBPB3.1) and SB (qSB3.1) (Figure 2). For the DTH trait, the Jupiter allele had an increasing effect as indicated by the negative additive values shown in Table 3.

### 2.4. Non-Parametric Analysis of the Markers Associated with BPB, SB, and DTH

Most QTL mapping methods assume that phenotypes follow a normal distribution within a population, however, the phenotypes of interest in this study were mostly not normally distributed (Figure 1) based on significant *p*-values (≤0.05) from a Shapiro–Wilk test (not shown). The most adopted method in dealing with this case was using a non-parametric analysis. Kruskal–Wallis’ test of markers was conducted to show association with the traits of interest.

Kruskal–Wallis’ test indicated that markers located in chromosome 3 in the genomic position interval from 0–13 cM were significantly associated with BPB, SB, and DTH. These markers also have significant LOD scores ranging from 5.62–18.2 (Table 4).

Markers associated with these significant positions were evaluated based on their effect on the genotypes and showed significant changes relative to alleles of Trenasse (AA) and Jupiter (BB). The associated marker for the significant position specifically for SB, BPB, and DTH was SNP225. The genotype of Jupiter (BB) had a decreasing effect on BPB and SB but an increasing effect on DTH (Appendix A).

### 2.5. Analysis of the Genes within the Major QTL Overlapping for SB, BPB, and DTH

The major QTL in the upper arm of chromosome 3, which is flanked by the markers SNP225 and SNP91 (Table 3) and associated with SB, BPB, and DTH as qSB3.1, qBPB3.1, and qDTH3.1, respectively, was subjected to singular enrichment analysis. The total number of annotated genes in that QTL region was 201, which could be expressed as 42 GO terms composed of 17 GO terms in the biological process category, 9 for molecular functions, and 16 for cellular components (Appendix A). While the majority of the 201 proteins expressed from the QTL region were predicted to be located in the chloroplast (69) and the nucleus (62), 41 and 46 proteins of them were predicted to be in the cytosol and the mitochondria, and 22 and 24 proteins in the plasma membrane and the extracellular space, respectively (Appendix A). In addition, a small number of proteins were putatively localized in the vacuole, the peroxisome, and the endoplasmic reticulum (Appendix A).

Functional annotation revealed that, from 201 genes found in the QTL, three genes were putatively related to pathogen recognition, jasmonic acid biosynthesis, and flowering, respectively (Table 5). The gene LOC_Os03g04110 encodes a putative receptor-like protein for a chitin elicitor, suggesting that this gene is involved in pathogen recognition. LOC_Os03g03810 encodes an antimicrobial peptide, which may be involved in the jasmonic acid (JA)-dependent defense response upon pathogen infection. The third gene, LOCOs03g03070, encodes a putative transcription factor for inducing short/long-day promotion of flowering (Table 5). These genes are within 1.0 to 1.9 Mb of chromosome 3.

### 2.6. QTL-Seq Analysis Using Whole-Genome Sequence Data of Selected RILs

For QTL-seq analysis, 15 RILs were selected primarily based on the BPB scores in both 2012 and 2013 for 8 BPB-resistant and 7 BPB-susceptible RILs. These two bulks representing contrasting BPB phenotypes showed the same contrasting pattern for the SB scores in 2012 and 2014, and for the DTH trait in 2012 and 2013 except one disease-resistant RIL showing an early DTH phenotype (Appendix A). High-throughput sequencing of 15 selected RILs and the parents generated 24.67 to 84.63 million reads per line, with a read mapping ratio ranging from 38.79 to 98.89%. This wide range of mapping rates (38.79–98.98%) was due to the low-quality reads of some samples and the stringent QC parameter (Q ≥ 25), which resulted in a large discrepancy between total reads and mapped reads (e.g., TJ-RIL173 and TJ-RIL255) (Appendix A). The coverage ranged from 20.09 to 49.92 per line (Appendix A). Individual whole-genome sequences of the selected RILs were combined based primarily on the BPB trait to generate BAM files of the resistant and susceptible bulks, which were composed of eight resistant and seven susceptible RILs, respectively (Appendix A). As mentioned above, the two bulks also represent contrasting phenotypes in the SB and DTH, except one RIL (TJ-RIL40) showing an early DTH phenotype in the resistant bulk (Appendix A). Thus, in this QTL-seq analysis, we considered that the contrasting trait of the two bulks represents not only the BPB trait but also the SB and DTH traits because they are highly correlated among the entire RIL population (Table 2) and the selected RILs of two bulks (Appendix A). The total number of reads for the resistant bulk was 255.69 million reads with 49.48 X average coverage, while the susceptible bulk had 226.82 million reads with 49.61 × average coverage (Appendix A).

Our QTL-seq analysis with the whole-genome sequence data of the 15 selected RILs, with the genotype of Trenasse as the reference, identified eight QTLs with an average interval of 2.01 Mb (Figure 3 and Table 6). The first QTL mapped by QTL-seq overlapped with the major QTL detected by the QTL linkage analysis in the 1.0 to 1.9 Mb region of chromosome 3. Six of the QTLs from chromosomes 1, 2, 3, 9, and 11 contained the Jupiter allele, while the two QTLs from chromosomes 3 and 8 were Trenasse alleles (Table 6). There were ~2500 (putative) genes harbored within the eight intervals. Among these, nine genes were annotated to be involved in plant defense, signaling, and flowering (Table 7). These genes were predicted to be involved in pathogen recognition, jasmonic acid biosynthesis and signaling, salicylic acid signaling, and flowering (Table 7). Three out of the nine genes were the same candidate genes identified by the QTL linkage analysis: LOC_Os03g04110, LOC_Os03g03810, and LOC_Os03g03070. However, only four genes of the nine candidate genes contained significant polymorphism between Jupiter and Trenasse, which were LOC_Os01g56200, LOC_Os03g03070, LOC_Os08g35740, and LOC_Os11g05480 (Table 8).

### 2.7. Marker Test for the Parents and the Selected RILs for the Resistant and Susceptible Bulks

Genome sequences of the parents and the individual RILs comprising the resistant and susceptible bulks were analyzed based on the four candidate genes showing polymorphisms between Jupiter and Trenasse (LOC_Os01g56200, LOC_Os03g03070, LOC_Os08g35740, and LOC_Os11g05480) (Table 8). Based on the sequence data, we identified six polymorphic markers on chromosome 1, seven markers for chromosome 8, one marker for chromosome 11, and three markers for chromosome 3 (Figure 4). For LOC_Os01g56200, five out of eight resistant RILs contained a Jupiter allele, while five out of seven susceptible RILs had the Trenasse allele (Figure 4). For LOC_Os11g05480, five resistant RILs had a Jupiter allele, while six susceptible RILs had a Trenasse allele (Figure 4). For LOC_Os03g03070, the majority of the resistant lines (seven out of eight RILs) contained the Jupiter allele, while the majority of the susceptible lines (six out of seven RILs) had the Trenasse allele (Figure 4). However, for LOC_Os08g35740, the reverse was observed, where more than half of the resistant and susceptible RILs had the Trenasse allele (five out of eight RILs) and the Jupiter allele (four out of seven RILs), respectively (Figure 4). Out of the seventeen polymorphisms, five were annotated to have a moderate to modifier effect on the phenotype due to non-synonymous mutation leading to a change in amino acid. The significant variants consisted of four SNPs and one deletion among the four candidate genes (Table 8). The effect impact categories in the SnpEff program (high, moderate, low, modifier) were used to determine the putative impact or deleteriousness of the variant on genes. The genotype calls of individuals comprising the resistant and susceptible bulks revealed that both parents contributed alleles for resistance. The majority (14 out of 15) of the individual RILs were homozygous for the seventeen markers targeting the four candidate genes.

## 3. Discussion

While there are more than 200 QTLs reported for the SB disease in rice [29], QTLs associated with BPB have not been studied much. Our literature search found only two studies conducted for identifying genomic regions that may confer resistance to BPB. Pinson et al. in 2010 [20] identified 12 QTLs associated with BPB resistance from a recombinant inbred line population developed from Lemont and TeQing. In 2013, Mizobuchi et al. [21] identified a major QTL in chromosome 1 that confers BPB resistance derived from Kele, an indica variety. Up to this point, no further or follow-up studies have been reported to validate the results of these findings nor identified genes that are responsible for BPB resistance.

In this study, we identified a major QTL on the upper arm of chromosome 3 from both linkage analysis and QTL-seq, which was simultaneously associated with the trait in BPB, SB, and days to heading (*qBPB3.1*, *qSB3.1*, and *qDTH3.1*). This finding was consistent with previous reports that the days to heading trait affected the resistance of rice varieties to SB and BPB [20,45]. This was also corroborated with the correlation observed in the RIL population, where early-heading RILs tend to show higher disease severity compared to the late-heading RILs, besides the fact that the late-heading parent Jupiter is more resistant to BPB and SB than the early-heading parent Trenasse. The major QTL on chromosome 3 identified in this study also overlaps with a QTL identified by Pinson et al. in 2010 [20] from another RIL population developed from Lemont and TeQing, which was the most statistically significant QTL for BPB and SB resistance. This region also overlaps with the previously mapped QTLs for SB, bacterial leaf blight (BLB), and days to heading [30,46].

Further investigation of this major QTL identified three candidate genes that might confer resistance to both BPB and SB, which were *LOC_Os03g03070* (encoding a MADS-box protein [44], *LOC_Os03g03810* (encoding an antimicrobial peptide [43]), and *LOC_Os03g04110* (encoding a receptor-like protein [41,42]). However, polymorphism between the parent varieties, Trenasse and Jupiter, was found only in *LOC_Os03g03070*. The polymorphic DNA sequence variations in that candidate gene are a SNP in an intron and a small deletion in the 3-prime UTR (Table 8), which might have modifier effects. *LOC_Os03g03070* was reported to be primarily involved in the short-day or long-day promotion of flowering [44]. As Pinson et al. suggested [20], we could speculate that the observed correlation between flowering time and disease phenotypes is probably because of conducive environmental or physiological conditions coinciding at the time of the infection process. This notion is consistent with the result of our analyses in this study, in which the early-heading RILs tend to show higher disease severity compared to the late-heading ones.

We also performed QTL-seq analysis to complement the QTL linkage analysis in this study, using the whole-genome sequence data of 15 selected RILs. Although we performed QTL-seq analysis with only eight resistant and seven susceptible RILs for the resistant and susceptible bulks, it was adequate to detect major loci involved in resistance. All the eight RILs of ‘the resistant bulk’ were more resistant to both BPB and SB than the seven RILs of ‘the susceptible bulk’, but one RIL (TJ-RIL40) in the resistant bulk showed an early-heading phenotype. Accordingly, all the BPB-susceptible RILs were also SB susceptible and early heading. As these three traits were highly correlated among the selected RILs, we consider the genomic regions detected by QTL-seq analysis to be associated with BPB, SB, and days to heading. The coverage of sequence data for each parent and bulk used in this study (20.1–49.9X) was within the range used for other QTL-seq studies previously reported (6–80X) [39,47,48].

Through the QTL-seq analysis, we could detect eight genomic regions in chromosomes 1, 2, 3, 8, 9, and 11. Interestingly, the QTL mapped on chromosome 1 overlapped with the QTL (*RBG2*) which was previously detected using a backcrossed inbred line population derived from the rice varieties Hitomebore and Kele [21]. *RBG2* was a 502 kb interval flanked by markers RM11725 and RM11727. This locus was later fine-mapped to RM1216 and RM11727 on chromosome 1 at the position of 31.7–32.2 Mb [46], which overlapped with the QTL mapped by QTL-seq in our study (Figure 3). Nevertheless, Mizobuchi et al. [21,49] did not find any predicted genes for disease resistance, such as nucleotide-binding-site–leucine-rich-repeat (NBS-LRR) genes. In this study, however, we could identify a candidate gene from the genomic position in chromosome 1 with a significant variation between Trenasse and Jupiter through expanding the region detected by the QTL-seq, which is *LOC_Os01g56200* encoding an NPR1-like protein. *NPR1* is an essential component for the salicylic acid signaling and the systemic and induced acquired resistance of the model plant, *Arabidopsis thaliana* [50]. *NPR1* acts as the master regulator for the plant defense signaling network, mediating crosstalk between the salicylic acid (SA)-dependent and the jasmonic acid/ethylene-dependent defense pathways [51]. This candidate gene in the locus contains a SNP between Trenasse and Jupiter for a single amino acid change between methionine and leucine (Table 8), which was predicted to have a moderate effect on the phenotype and most of the resistant RILs (five out of eight RILs) contain resistant allele (Jupiter) for this variant (Figure 4).

As discussed earlier in a previous paragraph, the region detected by the QTL-seq on chromosome 3 was within the major QTL (*qBPB3.1*/*qSB3.1*/*qDTH3.1*) found by the QTL linkage analysis, flanked by SNP225-SNP91. The region detected by the QTL-seq was smaller compared with the region detected by the linkage analysis (1.0 vs. 1.3 Mb). The three candidate genes within this region were discussed above.

Aside from chromosomes 1 and 3, new QTLs were mapped by the QTL-seq analysis on chromosomes 2, 8, 9, and 11 for BPB and SB. In these newly reported regions, five candidate genes were identified (Table 7). However, both parents only showed significant polymorphism in two genes, which would be the most viable candidates for resistance expression. These genes are *LOC_Os08g35740* (*OPEN GLUME1*) and *LOC_Os11g05480* (*b-ZIP TRANSCRIPTION FACTOR 79*). *LOC_Os08g35740* and *LOC_Os11g05480* are likely involved in jasmonic acid biosynthesis and salicylic acid signaling, respectively. These two genes also contain significant variations between Trenasse and Jupiter. The G to C conversion in *LOC_Os08g35740* results in a downstream gene variant predicted to have a modifier effect on the phenotype, while the conversion from A to G in *LOC_Os11g05480* was predicted to have a modifier effect (Table 8).

Conclusively, we could identify candidate rice genes involved in BPB and SB, as well as days to heading, through QTL linkage analysis and QTL-seq, although further genetic study is needed to validate the functions of those candidate genes. The genetic and statistical correlation observed with the phenotypic traits in BPB, SB, and days to heading from this RIL population made it difficult to evaluate the BPB and SB resistances of RILs without epistatic effect or linkage drag by loci for the days to heading trait. Utilizing a population showing a wide range of disease traits without significant difference in the days to heading trait would be ideal to identify QTLs solely associated with BPB and SB traits. Currently, we are conducting an additional genetic study with another set of Louisiana rice varieties, Bengal and Jupiter, which show contrasting phenotypes in BPB but similar traits in flowering time. Nevertheless, we expect that the information gained from this study could be used to develop more reliable molecular markers for the breeding of disease-resistant and/or early-maturing rice.

## 4. Materials and Methods

### 4.1. Generation of the RIL Mapping Population

A recombinant inbred line (RIL) population was developed from a cross between Jupiter, a medium-grain cultivar moderately resistant to bacterial panicle blight and sheath blight [40,52], and Trenasse, a long-grain cultivar susceptible to both diseases [53], through single seed descent. Seeds from a single panicle of each F2 plant were grown as a separate line in each generation. The RIL populations of F5, F6, and F7 generations composed of 286 to 300 RILs were used for their phenotypes in BPB and SB, as well as additional traits, between 2012 and 2014, and the 2016 growing seasons, respectively. Data collected in 2016 were not used in this study due to the extremely low disease pressure in that season.

### 4.2. Evaluation of Individual RILs in Their Phenotypes in BPB, SB, and other Traits

RILs and their parent lines were grown in the plots at the H. Rouse Caffey Rice Research Station (HRCRRS) (Rayne, Louisiana, USA) with two replications. Rice seeds were planted in the field using a Hege seed drill in mid-March to mid-April depending on weather conditions. Rice plants were managed according to the standard protocol of HRCRRS [54]. Each replication was a row contained approximately 15 to 20 plants. Phenotypic evaluation for BPB and SB, as well as other important agronomic traits (e.g., plant height and days to heading) was conducted for 286 to 300 RILs for each disease in 2012 and 2013. In 2014, only sheath blight was assessed with two replications.

Pathogen inoculation to evaluate the phenotypes in BPB and SB was conducted following the methods previously described [54] with minor modifications. Briefly, for BPB, *B. glumae* 336gr-1, a virulent strain of the pathogen [55,56], was grown overnight on LB agar at 37C, and subsequently resuspended in 10 mM MgCl2. Then, the bacterial suspension adjusted to a concentration of ~1 × 10^8^ CFU/mL (OD600 = 0.2) was sprayed onto rice plants at an early-heading stage (~30% to 50% heading) for inoculation using a 1 L hand sprayer. To reduce the bias due to unsynchronized flowering, inoculation was conducted four times with 2–4-day intervals, and the BPB phenotypes of RILs were evaluated 7 to 10 days after the last inoculation. For SB, the fungal pathogen *Rhizoctonia solani* was inoculated onto rice plants at an active tillering stage. The inoculum was prepared using a mixture of rice grains and hulls (1:2. *v/v*). The sterilized mixture in a 2 L flask was inoculated with ~16 cm^2^ of PDA plugs containing 5- to 7-day old *R. solani* mycelia followed by incubation at 25 °C in the dark for 10 days. After 10 days, this inoculated grain–hull mixture was mixed with a larger volume of an additional sterilized rice grain–hull mixture (1:2 *v*/*v*), which was then spread uniformly over a clean brown paper and covered with a clean plastic sheet at room temperature. The resulting grain–hull mixture was used as the inoculum after 24 h of incubation in the dark. This inoculum was applied to the rice plants by hand sprinkling at an early vegetative (tillering) stage, and the SB phenotypes of the RILs were evaluated 20 to 30 days after inoculation at the milk stage of rice.

For evaluation of BPB, symptom development was scored according to the visual percentage of discoloration of the panicles based on a 0 to 9 scale: 0 for no symptoms and 9 for more than 90% infected panicle area [6]. For SB, disease symptoms were evaluated during the milk stage based on a 0 to 9 scale following the standard proposed by the International Rice Research Institute [57]: 0 for no symptoms and 9 for more than 90% symptomatic area or collapsed plants. The days to heading trait was determined based on the number of days after planting when ~50% of the plants in the plot showed emerging panicles.

### 4.3. Genotyping of the RIL Population

Total genomic DNA was isolated for the RILs and the parents, Jupiter and Trenasse, using the CTAB method [58]. The concentration of DNA in each sample was estimated with an ND-1000 spectrophotometer (Thermo Fisher Scientific, Wilmington). The DNA concentration of all samples was adjusted to a final concentration of 25 ng/µL for PCR amplification. Genotyping of the RILs was conducted in the Molecular Marker Laboratory at the H. Rouse Caffey Rice Research Station, using 135 SNP markers designed for Kompetitive Allele Specific PCR (KASP).

### 4.4. Statistical Analyses of Phenotypic Traits

Means of the traits for individual parents and RILs in each year as well as across years were calculated. Analysis of variance and broad sense heritability were calculated using GLM in R [59]. The correlations among the phenotypic traits were calculated using Pearson’s correlation coefficients for each year. The heritability of each trait across years was calculated using the formula for broad sense heritability [60]. Heritability (h2) is the proportion of observed phenotypic variations due to genetic differences. A greater heritability value in a population suggests that the phenotypic variation is due to genetics while low heritability in a population suggests high environmental influence in the variation observed within the population.

### 4.5. QTL Linkage Analysis

The genotypic data of the RILs were used to construct the linkage map by using the physical positions of the SNPs. If two adjacent markers contained the same allele, the chromosome segment was assumed to be composed entirely of that marker genotype. When two adjacent markers exhibited different alleles, the interval was equally divided between the two markers. The possibility of double recombination within the interval was disregarded. The genotypic data were used for the generation of a linkage map using QTL IciMapping software v.4.2.53 [61] (www.isbreeding.net/software, accessed on 5 April 2022), and the linkage map combined with the phenotypic data was used to identify the QTLs associated with the traits. The QTL IciMapping software was also used for QTL mapping. Inclusive composite interval mapping for additive QTL (ICIM-ADD) was used to identify QTLs at the logarithm of the odds (LOD) threshold of 2.5. The position for QTLs and their effects were also estimated. The information of the genes within selected QTL intervals was obtained from the Rice Genome Annotation Project Database Release 7 (http://rice.plantbiology.uga.edu/, accessed on 5 April 2022). For the non-parametric analysis of QTLs, the R/qtl v.1.48-1 package [62] was used with a non-parametric model set combined with 1000 permutations. Single gene enrichment analysis was performed with the genes in the selected genomic regions using the plant set enrichment analysis tool kit [63] at the 0.05 false discovery rate (FDR), and significant GO terms and genes putatively associated with BPB, SB, and days to 50% heading were identified.

### 4.6. QTL-Seq Analysis of the Resistant and Susceptible Bulks

Eight BPB-resistant RILs and seven BPB-susceptible RILs were selected as the components of the resistant and susceptible bulks, respectively, based on the BPB phenotypes in 2012 and 2013. The RILs selected for each bulk also exhibited similar patterns in the SB phenotype. Thus, each bulk was considered to represent the resistance or the susceptibility to both BPB and SB. One-week-old rice seedlings of the 15 selected RILs were used to isolate total genomic DNA using a DNeasy Plant Mini Kit (Qiagen, Valencia, CA, USA). The concentration of the DNA samples was measured using a Nano Spectrophotometer (Nano Drop, Wilmington, DE, USA). For the high-throughput DNA sequencing, the DNA samples were sent to the Virginia Bioinformatics Institute (VBI) Genomics Lab (now Genomics Sequencing Center) at Virginia Tech. The Nextera DNA library preparation kit was used to develop genomic DNA libraries for paired-end sequencing to generate 100-base long reads. The quality of sequence reads of the RILs was examined using FastaQC [64]. Trimming of the adapter was implemented using Trimommatic using default parameters [65] and mapped to the International Rice Genome Sequence Project (IRGSP) pseudomolecules version 7 of the reference genome of the Japonica cultivar Nipponbare, using BWA-MEM with its default parameter [66]. SAM files were sorted and converted to BAM using Picard tools. The QTL-seq R package (version 2.2.2) was used to detect QTLs [67]. The input SNP file was filtered based on average coverage per sample, such that each SNP had a read depth of no less than 45 for each bulk. The cutoff was determined by exploring the data with read depth histograms and following the recommended QTL-seq guidelines. The SNP-index was calculated for all SNPs in the two pools of mixed samples, with the genotype of Trenasse as the reference [68]. During the calculation, SNPs with a SNP-index <0.3 or >0.7 were filtered to denote a co-segregation of certain genotypes in the two pools. A 200 kb sliding window with a 10 kb increment was applied to slide across the genome. The average value of the SNP-index was calculated in each window. The ΔSNP-index of Trenasse (from −1 to +1) was calculated using the SNP-index of the resistance pool minus the SNP-index of the susceptible pool and plotted along the 12 chromosomes of *O. sativa* to present the signals detected by the QTL-seq analysis.

### 4.7. Comparative Sequence Analysis

Variant calling and generation of VCF were conducted using GATK Haplotypecaller [69]. VCFs were comparatively analyzed by using vcftools [70]. Variants and their effects on different chromosomal regions were annotated using SnpEff with the available pre-built database for *Oryza sativa* within the program [71].

## 5. Conclusions

We conducted a genetic study for the quantitative disease resistance of rice to BPB and SB through QTL mapping and bulk segregant analysis coupled with whole-genome sequencing of rice genomes. By utilizing the RIL population derived from two Louisiana rice cultivars, Jupiter and Trenasse, this study revealed candidate loci in the genome of both parents that could explain the variability in the resistance phenotype. One major QTL on chromosome 3 was identified from our QTL linkage analysis and QTL-seq, which was associated with the traits in BPB, SB, and DTH. This locus was overlapped with a QTL previously identified for BPB and SB. The close linkage between the resistance to BPB and SB and the days to heading trait within this major QTL made it difficult to determine the epistatic or autonomous effect of this QTL on disease phenotypes. The QTL-seq analysis in this study revealed another major QTL on chromosome 1, which was also overlapped with a previously identified QTL associated with BPB resistance, as well as several minor QTLs in other parts of the rice genome. Candidate genes present in the major QTLs identified in this study are known or predicted to function in flowering or defense. To the best of our knowledge, this study was the first attempt to use both QTL linkage analysis and QTL-seq analysis to identify QTLs for BPB resistance. The genetic information and the rice RIL population gained from this study will be a useful resource for breeding programs to develop disease-resistant lines.

## Figures and Tables

**Figure 1 plants-12-00559-f001:**
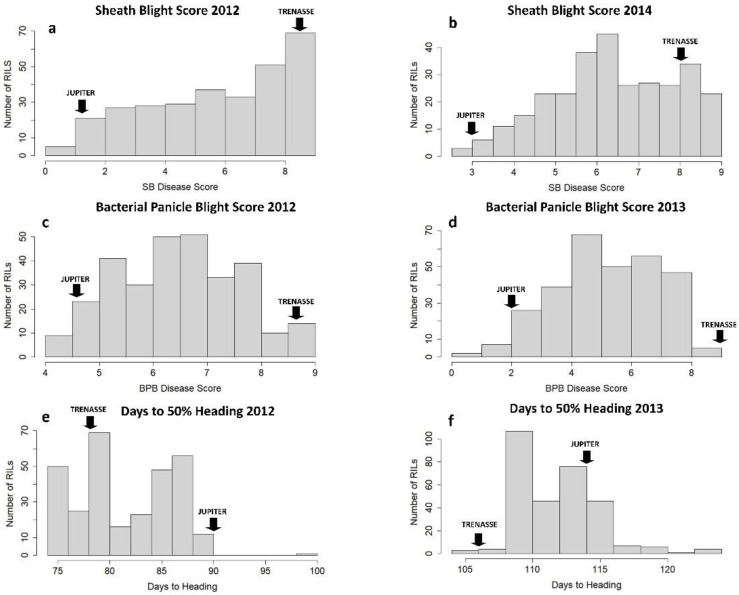
Phenotypic frequency distribution of the RIL population for three traits, sheath blight (**a**,**b**), bacterial panicle blight (**c**,**d**), and days to 50% heading (**e**,**f**), in two different years. ‘JUPITER’ and ‘TRENASSE’ indicate the mean phenotypic values for the parents.

**Figure 2 plants-12-00559-f002:**
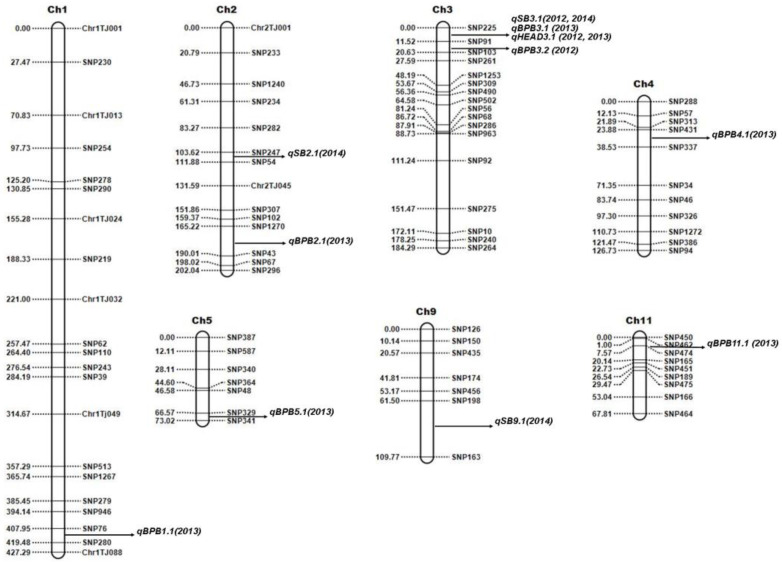
Genetic linkage map of the Trenasse/Jupiter RIL population showing the positions of major and minor QTLs for bacterial panicle blight, sheath blight (SB), and days to 50% heading (DTH). QTLs were detected by inclusive composite interval mapping using QTL IciMapping software v. 4.2.53.

**Figure 3 plants-12-00559-f003:**
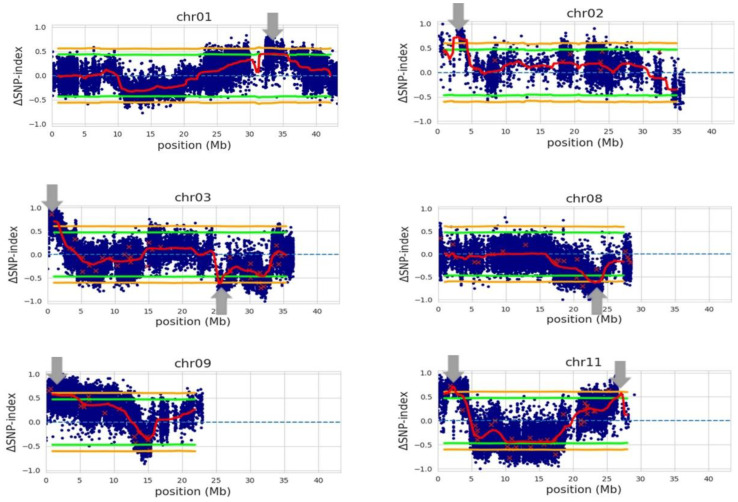
QTLs (indicated by gray arrows) associated with bacterial panicle blight (BPB), sheath blight (SB), and days to 50% heading (DTH) using Trenasse as the consensus reference genome. The blue dots correspond to individual SNP variants between two bulks, which represent ‘BPB/SB-susceptible and early DTH’ and ‘BPB/SB-resistant and late DTH’, respectively. The red ‘x’s indicate variants with low depth. The red lines represent the mean for ΔSNP-indices. The green and orange lines are the 95% and 99% confidence intervals for the regions, respectively.

**Figure 4 plants-12-00559-f004:**
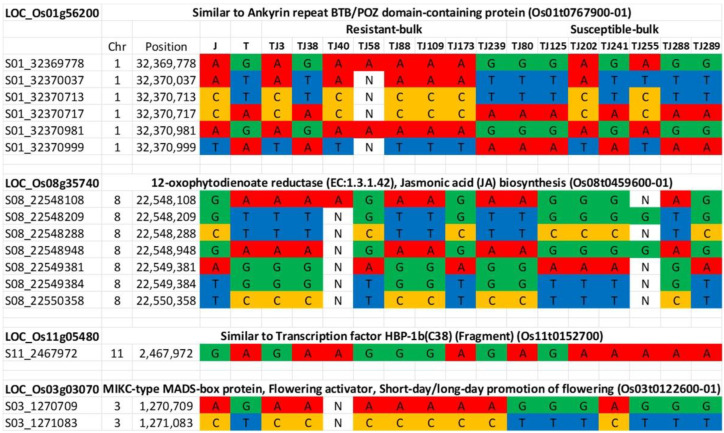
Genotypes across sixteen markers for the parents (Jupiter ‘J’ and Trenasse ‘T’) and individuals constituting the susceptible (7 RILs) and resistant (8 RILs) bulks. ‘N’ represents missing genotypes.

**Table 1 plants-12-00559-t001:** Mean values and ranges of phenotypic traits of the parents and the recombinant inbred lines (RILs) for bacterial panicle blight (BPB), sheath blight (SB), and days to 50% heading (DTH).

Trait Name	Mean Trenasse/Jupiter ^a^	RIL Mean	RIL Range	Std. Dev. ^b^	RIL F-Values ^c^	Heritability ^d^
SB2012 ^e^	8.9/ 1.6 ***	5.9	0–9	2.4	11.30 **	0.4888
SB2014 ^e^	8.0/ 3.0 ***	6.6	2.5–9	1.51	2.67 **	
BPB2012 ^e^	8.7/ 4.4 ***	6.7	4–9	1.15	2.31 **	0.2366
BPB2013 ^e^	8.6/ 1.9 ***	5.4	0.5–8.5	1.68	6.29 **	
DTH2012 ^f^	79.0/ 89.0 ***	81.9	74–99.5	4.58	9.79 **	0.7136
DTH2013 ^f^	108.0/ 114.0 ***	112.5	105–124	3.12	6.45 **	

Trait names: SB2012, sheath blight disease index for 2012; SB2014, sheath blight disease index for 2012; BPB2012, bacterial panicle blight disease index for 2012; BPB2013, bacterial panicle blight disease index for 2013; DTH2012, days to heading for 2012; DTH2013, days to heading for 2013. ^a^: *T*-test between the means of Trenasse and Jupiter; ^b^: standard deviations of RILs; ^c^: ANOVA of RILs; ^d^: broad sense heritability computed on family mean basis (across years); ^e^: numeric values indicate the disease scores in a 0-to−9 scale; ^f^: numeric values indicate the number of days to 50% heading after planting; ** significant at *p* < 0.01; *** significant at *p* < 0.001.

**Table 2 plants-12-00559-t002:** Pearson correlation matrix of bacterial panicle blight (BPB), sheath blight (SB), and days to 50% heading (DTH) traits in RILs.

	SB 2012	SB 2014	BPB 2012	BPB 2013	DTH 2012	DTH 2013
**SB 2012**	1					
**SB 2014**	0.72 *	1				
**BPB 2012**	0.61	0.55	1			
**BPB 2013**	0.84 **	0.63	0.43	1		
**DTH 2012**	−0.99 *	−0.75 *	−0.63 *	−0.85 **	1	
**DTH 2013**	−0.92 **	−0.80 *	−0.65 *	−0.89 **	0.91 **	1

* Significant at *p* < 0.05. ** Significant at *p* < 0.01.

**Table 3 plants-12-00559-t003:** Additive QTLs detected for the traits related to bacterial panicle blight (BPB), sheath blight (SB), and days to 50% heading (DTH) in the RILs by inclusive composite interval mapping.

Trait	QTL	Ch	Position (cM)	Marker Interval	Marker Position (Mb)	LOD	PVE (%)	Additive Effect	Published QTLs
SB 2012	* **qSB3.1** *	3	5	SNP225-SNP91	0.66–1.95	19.37	26.34	1.3881	[20]
SB 2014	* **qSB2.1** *	2	109	SNP247-SNP54	21.76–24.73	4.62	7.74	0.4168	
	* **qSB3.1** *	3	1	SNP225-SNP91	0.66–1.95	3.63	5.49	0.3501	[20]
	* **qSB9.1** *	9	109	SNP198-SNP163	15.62–22.74	2.9	4.32	0.3105	
BPB 2012	* **qBPB3.2** *	3	13	SNP91-SNP103	1.95–3.39	4.24	6.04	0.303	
BPB 2013	* **qBPB11.1** *	11	5	SNP462-SNP474	4.13–5.37	3.42	4.97	−0.2732	
	* **qBPB1.1** *	1	416	SNP76-SNP280	40.13–42.32	2.8	3.26	0.353	
	* **qBPB2.1** *	2	182	SNP1270-SNP43	29.79–34.37	2.91	4.72	0.3889	
	* **qBPB3.1** *	3	2	SNP225-SNP91	0.66–1.95	16.64	19.82	0.7946	[20]
	* **qBPB4.1** *	4	29	SNP431-SNP337	11.87–15.98	2.8	3.56	−0.3369	
	* **qBPB5.1** *	5	69	SNP329-SNP341	19.44–20.88	3.88	4.28	0.3736	
DTH 2012	* **qHEAD3.1** *	3	5	SNP225-SNP91	0.66–1.95	20.22	25.99	−2.7605	
DTH 2013	* **qHEAD3.1** *	3	2	SNP225-SNP91	0.66–1.95	15.2	22.01	−1.5383	

**Table 4 plants-12-00559-t004:** Markers associated with bacterial panicle blight (BPB), sheath blight (SB), and days to 50% heading (DTH).

Trait	Position ^a^	Marker	Kruskal–Wallis Test *p*-Value	LOD Score
SB 2012	Ch 3 (Loc 5)	SNP225	0.000 **	18.2
SB 2014	Ch 2 (Loc 110)	SNP54	0.000 **	5.16
	Ch 3 (Loc 0)	SNP225	0.000 **	5.62
BPB 2012	Ch 3 (Loc 13)	SNP91	0.006 **	3.99
BPB 2013	Ch 2 (Loc 176)	SNP1270	0.024 **	3.31
	Ch 3 (Loc 3)	SNP225	0.000 **	15.86
	Ch 5 (Loc 60)	SNP329	0.009 **	3.67
DTH 2012	Ch 3 (Loc 4)	SNP225	0.000 **	17.4
DTH 2103	Ch3 (Loc 2)	SNP225	0.000 **	15.6

^a^: Chromosome (Ch), genetic location associated with a trait (Loc #). ** Significant at *p* < 0.01.

**Table 5 plants-12-00559-t005:** List of candidate genes identified from the major QTL on chromosome 3 overlapping for SB, BPB, and DTH.

MSU Rice Annotation	Pathway Identifier	Pathway Name	Gene Name	Description	Reference
*LOC_Os03g04110*	R-OSA-9612650	Responses to stimuli: biotic stimuli and stresses	Chitin elicitor-binding protein	Lysine motif (LysM) receptor-like protein (RLP), chitin oligosaccharide elicitor-binding protein, perception and transduction of chitin elicitor signal for defense responses	[41,42]
	R-OSA-9611432	Recognition of fungal and bacterial pathogens and immunity response			
*LOC_Os03g03810*	R-OSA-6787011	Jasmonic acid signaling	Defensin-like 8	Defensin, plant antimicrobial peptide, pathogen defense	[43]
*LOC_Os03g03070*	R-OSA-8934036	Long-day regulated expression of florigens	MADS-box gene 50	MIKC-type MADS-box protein, flowering activator, short-day/long-day promotion of flowering (Os03t0122600-01); transcription factor, MADS-box domain-containing protein	[44]

**Table 6 plants-12-00559-t006:** QTLs (*p* < 0.05) detected by QTL-seq associated with the resistance to BPB, SB and DTH using Trenasse as the reference genome.

Chromosome	Start (bp)	End (bp)	Interval (bp)	Mean ∆SNP-Index	Allele in R-Bulk
1	31,674,255	34,217,945	2,543,690	0.598	Jupiter
2	2,432,622	4,317,628	1,885,006	0.713	Jupiter
3	1,002,165	1,997,968	995,803	0.693	Jupiter
3	25,310,338	25,994,426	684,088	−0.668	Trenasse
8	21,700,098	24,288,071	2,587,973	−0.652	Trenasse
9	1,002,205	4,792,809	3,790,604	0.611	Jupiter
11	1,003,453	3,799,429	2,795,976	0.666	Jupiter
11	26,401,483	27,226,417	824,934	0.646	Jupiter

**Table 7 plants-12-00559-t007:** Candidate genes related to plant defense, signaling, and flowering revealed by QTL-seq analysis.

MSU Locus	Gene Name (Gene Symbol) in RAP-DB
*LOC_Os01g56200*	NPR1 HOMOLOG 2 *(NH2, OsNH2, OsNPR2, NPR2, DLN21, OsDLN21, PR2)*
*LOC_Os02g05510*	GATA TRANSCRIPTION FACTOR 17 *(GATA17, OsTIFY2a, OsCCT04, OsGATA17, OsGATA17b)*
*LOC_Os03g03070*	MADS BOX GENE 50 *(MADS50, OsMADS50, AGL20, SOC1, OsSOC1, RMADS208, DTH3, OsDTH3)*
*LOC_Os03g03810*	defensin 8 *(OsDEF8)*
*LOC_Os03g04110*	CHITIN ELICITOR BINDING PROTEIN *(CEBiP, OsCEBiP)*
*LOC_Os08g35740*	OPEN GLUME1 *(OG1, OPR7, OsOPR08-1, OsOPR13, OsOPR7)*
*LOC_Os11g05480*	b-ZIP TRANSCRIPTION FACTOR 79 *(BZIP79, OsbZIP79)*
*LOC_Os11g44600*	*Encoding a hypothetical protein involved in salicylic acid signaling* *
*LOC_Os11g44680*	*Not named but described as ‘Calmodulin binding protein-like family protein’*

* From a pathway analysis using Gramene (gramene.org).

**Table 8 plants-12-00559-t008:** Significant variants predicted to have impact on the resistance.

Genes	Variant (*p* < 0.05)	Type	Position	Change	Effect	Changes
*LOC_Os01g56200*	S01_32370037	SNP	32370037	A > T	Moderate	Missense variant; Met > Leu
*LOC_Os03g03070*	S03_1271083	SNP	1271083	C > T	Modifier	Intron variant
	S03_1270417	INDEL	1270417	CATAT > CAT	Modifier	3’ UTR variant
*LOC_Os08g35740*	S08_22548209	SNP	22548209	G > T	Modifier	Downstream gene variant
*LOC_Os11g05480*	S11_2467972	SNP	2467972	A > G	Modifier	Intron variant

## Data Availability

The NGS sequence data presented in this study will be openly available in the NCBI database (BioProject ID: PRJNA918354) once this manuscript is published.

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
