# Peer review of "Genetic Characterization of the Partial Disease Resistance of Rice to Bacterial Panicle Blight and Sheath Blight by Combined QTL Linkage and QTL-seq Analyses"

_plants, 2023, doi:10.3390/plants12030559_

Round 1

Reviewer 1 Report

The manuscript by Ontoy et al. reports QTL identification for bacterial panicle blight and sheath blight resistance by combined QTL linkage mapping and QTL-seq analyzes. They successfully identified a major QTL on chromosome 3, which was found to be a colocalized QTL for both disease resistance traits and for the day to heading (DTH) trait. Although the results of this study are interesting, I have a number of suggestions, which are listed below.

- In the introduction, please justify why BPB and SB need to be studied together.

- Also, please mention previous studies on QTL identification for BPB resistance in the third paragraph of Introduction.

- In the fourth paragraph of the introduction, please add the references for using QTL-seq for QTL identification, especially for disease resistance.

- In the last paragraph of the introduction, please mention that two approaches were used for QTL identification in the study.

- Table 1: Please provide details of the trait names in the Table description.

- In order to make the studied traits understandable to readers, please provide a figure showing the differences in disease resistance phenotypes among the parental lines.

- Please explain the difference in trait distribution between DTH2012 and DTH2013.

- Figure 1: Please add labels of subfigures a - f for each graph.

- Table 3: Please provide the physical position intervals in Mb for the flanking markers of each detected QTL. Also, please add a column for references for previously identified QTLs that were found to overlap with the QTLs identified in this study.

- Table 3 and Figure 2: Please check the summary in the Table and the position on the map for qBPH3.1 and qBPB 3.2. I found some inconsistencies.

- Figure 2: Please correct the QTL name on Chr.9. There is no QTL on chromosome 9 for BPB in Table 3.

- In section 2.3 of Results, second paragraph, line 5, "qSB3.1 was detected in both 2012 and 2013", please correct the years to 2012 and 2014.

- Please also provide a criterion for how a QTL can be considered a major QTL.

- In section 2.5 of Results, what criteria did the authors use to narrow the number of candidate genes, from 201 to 3 genes? 

- In section 2.6 of Results, what criteria were used to select 15 plants for QTL-seq? The selection was based on phenotypes from what year?

- Please provide a reason for the wide range of mapping rates (38.79 - 98.98%) among the 15 samples. 

- Again, it is not clear what criteria the authors used to prioritize candidate genes for each QTL identified by QTL-seq. 

- In the section 4.1 of Materials and Methods, the last sentence, were the experiments also performed in 2016? It was not included in the Results.

- In section 4.5 of Materials and Methods, please check the citation of reference #65 and #66. I think the reference for QTL-seq should be #66 (Sugihara et al. 2020).

Author Response

Reviewer 1

The manuscript by Ontoy et al. reports QTL identification for bacterial panicle blight and sheath blight resistance by combined QTL linkage mapping and QTL-seq analyzes. They successfully identified a major QTL on chromosome 3, which was found to be a colocalized QTL for both disease resistance traits and for the day to heading (DTH) trait. Although the results of this study are interesting, I have a number of suggestions, which are listed below.

- In the introduction, please justify why BPB and SB need to be studied together.

Response: This was purely based on the available genetic resource we had, specifically the donor parent, Jupiter. It was indicated in the methodology section that the donor parent, Jupiter, was moderately resistant to both BPB and SB. To emphasize that point, we added the following sentence to the introduction in the revision: “The donor parent, Jupiter, has been known to be both moderately resistant to BPB and SB [51]. Hence, it was important to determine common and unique genomic regions that confer the resistance to those diseases.” (Line 96 – 98 in the revision)

- Also, please mention previous studies on QTL identification for BPB resistance in the third paragraph of Introduction.

Response: A statement for previous QTL studies on BPB was incorporated in the second paragraph rather than in the third paragraph just to maintain each paragraph focusing exclusivity on each disease, respectively.

- In the fourth paragraph of the introduction, please add the references for using QTL-seq for QTL identification, especially for disease resistance.

Response: We could not find appropriate references to cite for previous QTL-seq studies on disease resistance. This could be due to our insufficient effort for literature search. Any suggested references from reviewers will be appreciated and included in the text.

- In the last paragraph of the introduction, please mention that two approaches were used for QTL identification in the study.

Response: A statement was inserted to indicate the two approaches used for QTL identification in this study. (Line 94 – 96 in the revision)

- Table 1: Please provide details of the trait names in the Table description.

Response: Trait names were explained as foot notes in the revised Table 1. (Line 125 – 127 in the revision)

- In order to make the studied traits understandable to readers, please provide a figure showing the differences in disease resistance phenotypes among the parental lines.

Response: Fig. S1 was newly added to present the disease phenotypes of the parents, Jupiter and Trenasse. Accordingly, previous Fig. S1, S2 and S3 were cited as S2, S3 and S4, respectively, in the revision. In addition, the following sentence was added. “Photo images that represent the disease phenotypes of Jupiter and Trenasse are presented in Fig. S1.” (Line 109-110 in the revision)  

- Please explain the difference in trait distribution between DTH2012 and DTH2013.

Response:  Following sentence “The large shift of DTH between 2012 and 2013 seasons was likely due to different environmental conditions (especially weather conditions) in the field” was added (Line 115-116 in the revision).

- Figure 1: Please add labels of subfigures a - f for each graph.

Response: Fig. 1 was modified as suggested. The figure legend was modified accordingly.  

- Table 3: Please provide the physical position intervals in Mb for the flanking markers of each detected QTL. Also, please add a column for references for previously identified QTLs that were found to overlap with the QTLs identified in this study.

Response: Physical position of flanking markers for each QTL and the reference for the overlapping QTL previously published were incorporated in Table 3.

- Table 3 and Figure 2: Please check the summary in the Table and the position on the map for qBPB3.1 and qBPB 3.2. I found some inconsistencies.

Response: Errors were corrected in Table 3, where qBPB3.1 for BPB2012 was changed to qBPB3.2 and qBPB3.2 for BPB2013 was changed to qBPB3.1. 

- Figure 2: Please correct the QTL name on Chr.9. There is no QTL on chromosome 9 for BPB in Table 3.

Response: Fig. 2 was corrected, changing qBPB5.1 to qSB9.1.

- In section 2.3 of Results, second paragraph, line 5, "qSB3.1 was detected in both 2012 and 2013", please correct the years to 2012 and 2014.

Response: The 2013 was changed to 2014 for SB. (Line 189 in the revision)

- Please also provide a criterion for how a QTL can be considered a major QTL.

Response: “QTL explaining less than 10.0% of the phenotypic variation was defined as minor QTL, whereas QTL with ≥ 10% was defined as major QTL” was inserted in the second paragraph of 2.3 section of Results. (Line 186 – 188 in the revision)

- In section 2.5 of Results, what criteria did the authors use to narrow the number of candidate genes, from 201 to 3 genes? 

Response: It was mentioned that through function annotation or in-silico analysis of those 201 genes, three were predicted to be directly involved in the defense and flowering traits. (Line 239 – 241 in the revision)

- In section 2.6 of Results, what criteria were used to select 15 plants for QTL-seq? The selection was based on phenotypes from what year?

Response:  To answer this question, the following sentences were added to the revision for providing this information: “For QTL-seq analysis, 15 RILs were selected primarily based on the BPB scores in both 2012 and 2013 for 8 BPB-resistant and 7 BPB-susceptible RILs. These two bulks representing contrasting BPB phenotypes showed the same contrasting pattern for the SB scores in 2012 and 2014, and for the DTH trait in 2012 and 2013 except one disease resistant RIL showing an early DTH phenotype (Table S2).” (Line 252 – 256 in the revision)      

- Please provide a reason for the wide range of mapping rates (38.79 - 98.98%) among the 15 samples. 

Response:  The following sentence was added, “This wide range of mapping rates (38.79 – 98.98%) was due to the low quality of reads of some samples and the stringent QC parameter (e.g., Q ≥ 25), which resulted in large discrepancy between total reads and mapped reads (e.g., TJ-RIL173 & TJ-RL225) (Table S1). (Line 258 – 261 in the revision)

- Again, it is not clear what criteria the authors used to prioritize candidate genes for each QTL identified by QTL-seq. 

Response: As we’ve mentioned earlier, prioritizing candidate genes was based on their predicted functions in defense or flowering through in silico analysis using available databases to annotate function of candidate genes. (Line 239 – 241 in the revision)

- In the section 4.1 of Materials and Methods, the last sentence, were the experiments also performed in 2016? It was not included in the Results.

Response: Though it was mentioned that evaluation was also done during 2016, due to some unavoidable circumstances (weather), disease pressure was not that high in the field providing difficulty in assessing in the disease’s response (no variations) of the parents and mapping population. Hence, 2016 data was not included in the analysis. This information is included in the revision. (Line 452 – 453 in the revision)  

- In section 4.5 of Materials and Methods, please check the citation of reference #65 and #66. I think the reference for QTL-seq should be #66 (Sugihara et al. 2020).

Response: Correction was made.

Reviewer 2 Report

Genetic Characterization of the Partial Disease Resistance of Rice to Bacterial Panicle Blight and Sheath Blight by Combined QTL Linkage and QTL-seq Analyses

Ontoy et al. made efforts to investigate the genetics of Bacterial Panicle Blight (BPB) and Sheath Blight (SB) by QTL linkage mapping coupled with QTLseq analysis. They identified a major effect locus at the start of Chromosome 3, among others for SB, BPB and days to heading.

The rationale for such a study is well justified, however the following limitations needs to be considered:

1.     The authors used progenies from quite an early generation (F4 & F5!) for screening the target phenotypes. One can expect a significant heterozygosity in such population and might eventually have effect on the overall phenotype. How was this addressed? Also, at which generation genotyping was done?

2.     It would be great if the authors can provide more details on how the inoculation was handled to account for the variation in flowering time. The observed linkage of resistance with flowering time might be because of the variation in the inflorescence development during inoculation?

3.     Another major concern is the analysis of candidate genes. This part is rather descriptive and the conclusions are not supported by the conducted experiments. For eg., the authors indicate alternate splicing as the mechanism of action of LOC_Os03g03070. However, no gene expression data is provided to support this conclusion. Apart from this, there are many other similar cases of speculations that are not verified. It is better to remove this part entirely or it is important to conduct relevant experiments. 

Author Response

Reviewer 2

Ontoy et al. made efforts to investigate the genetics of Bacterial Panicle Blight (BPB) and Sheath Blight (SB) by QTL linkage mapping coupled with QTLseq analysis. They identified a major effect locus at the start of Chromosome 3, among others for SB, BPB and days to heading.

The rationale for such a study is well justified, however the following limitations needs to be considered:

  1. The authors used progenies from quite an early generation (F4 & F5!) for screening the target phenotypes. One can expect a significant heterozygosity in such population and might eventually have effect on the overall phenotype. How was this addressed? Also, at which generation genotyping was done?

Response: Correction was made where initial population used for phenotyping was actually F5 followed by F6. (Line 447 in the revision).  A study simulated and suggested that the performance of QTL mapping with F4 population could be almost comparable to that with F6 or F7 populations given also a potential scenario where in using F4 population has that a large proportion of heterozygotes (Takuno S, Terauchi R, Innan H. The power of QTL mapping with RILs. PLoS One. 2012;7(10):e46545. doi: 10.1371/journal.pone.0046545. Epub 2012 Oct 9. PMID: 23056339; PMCID: PMC3467243). The genotypic data of this study was generated from the F6 population, and we found that the level of heterozygosity for the given KASP-markers (135 markers) was very low (> 5%).

  1. It would be great if the authors can provide more details on how the inoculation was handled to account for the variation in flowering time. The observed linkage of resistance with flowering time might be because of the variation in the inflorescence development during inoculation?

Response: It was mentioned in the methodology, specifically for BPB evaluation, since there was no synchronization of heading in a population to reduce the chance of escapes, at least four inoculations were done in 2-4 days interval. (Line 467 – 469 in the revision).  It is possible that timing and any physiological changes during inflorescence development are influencing the disease development, particularly the entry and spread of the pathogen.  

  1. Another major concern is the analysis of candidate genes. This part is rather descriptive and the conclusions are not supported by the conducted experiments. For eg., the authors indicate alternate splicing as the mechanism of action of LOC_Os03g03070. However, no gene expression data is provided to support this conclusion. Apart from this, there are many other similar cases of speculations that are not verified. It is better to remove this part entirely or it is important to conduct relevant experiments. 

Response: The major purpose of this study was to identify genomic regions contributing to the phenotypic variance observed within the given population. Any downstream analysis to characterize those regions were purely computational and predictive by utilizing the resources in public databases. Those variations observed in the “candidate genes” were merely predictive in nature based on in silico analysis. We strongly agree with the reviewer that molecular genetic study for these candidate genes should be followed to determine their functions in disease phenotypes. Nevertheless, we prefer keeping the parts that address putative functional mechanisms of candidate genes, including alternative splicing, to provide possible directions for future studies on rice disease resistance.

Round 2

Reviewer 1 Report

The authors have clearly addressed most of my comments and improved the manuscript. I have no further comments on the current version of the manuscript.

Author Response

Thank you very much for accepting our revised manuscript.

Reviewer 2 Report

I still feel that the candidate genes part, especially the underlying mechanism is over-speculative and primarily descriptive; not backed by any experimental evidence. There are several publically available gene expression resources in the case of Rice. Perhaps, the authors might consider performing a meta-expression analysis of the claimed 'candidate genes'. However, even then it is very much required to tone down this part.

Round 3

Reviewer 2 Report

I'm happy with the updates made by the authors in the current version of the manuscript.